# Case Report of a Unique Intra-Operative Finding in a Pediatric Distal Radius Non-Union: Does This Shed Light on Other Non-Unions or Malunions in Children?

**DOI:** 10.3390/children11010116

**Published:** 2024-01-17

**Authors:** Marius Negru, Adrian Emil Lazarescu, Corina Maria Stanciulescu, Liliana Catan, Calin Marius Popoiu, Eugen Sorin Boia

**Affiliations:** 1Department of Pediatric Surgery, Faculty of Medicine, “Victor Babes” University of Medicine and Pharmacy, 300041 Timisoara, Romania; marius.negru@umft.ro (M.N.); stanciulescu.corina@umft.ro (C.M.S.); mcpopoiu@umft.ro (C.M.P.); boia.eugen@umft.ro (E.S.B.); 2Luis Turcanu Emergency Children’s Hospital, 300011 Timisoara, Romania; 3Department of Anatomy, Faculty of Medicine, “Victor Babes” University of Medicine and Pharmacy, 300041 Timisoara, Romania; 42nd Clinic of Orthopaedics and Traumatology, Timisoara Emergency County Hospital, 300723 Timisoara, Romania; 5Teodor Sora Research Centre, UMFT, Department of Orthopaedics and Traumatology, 300041 Timisoara, Romania; 6Department of Rehabilitation, Faculty of Medicine, “Victor Babes” University of Medicine and Pharmacy, Physical Medicine and Rheumatology, 300041 Timisoara, Romania; catan.liliana@umft.ro

**Keywords:** pediatric trauma, radius non-union, traumatic tendon transposition, case report

## Abstract

Non-unions are quite rare in closed fractures in children. Most distal radius fractures require orthopedic reduction and conservative treatment with very good radiological and clinical/functional results. In case of unsatisfactory reduction, surgical treatment is necessary to correct significant displacement. Surgical treatment consists of closed reduction and percutaneous fixation using K-wires. If closed reduction is not possible, open reduction and fixation is mandatory. Generally, fixation is obtained using K-wires, in most cases, even if open reduction is necessary, rarely locking plates, especially in adolescents. The present paper presents a case of non-union that eventually required open reduction and plating. During surgery, however, it became evident that the cause for non-union was the traumatic transposition of the long extensor radialis tendon, through the fracture site to the volar side of the distal forearm. The movement of the carpus translated to constant mobility in the fracture site, leading to non-union and a continuous tendency towards anterior angulation of the distal fragment. The tendon was reduced to its anatomical position, the fracture was reduced, and fixed using a locking plate, and union was achieved with no complications. Traumatic transpositions of tendons should be considered in pediatric non-unions, and restoring anatomy is essential.

## 1. Introduction

Forearm fractures are the most common type of injury among all fractures in school-aged children, and most often conservative management with orthopedic reduction and cast immobilization leads to very good clinical outcomes [1]. Although reduction does not need to be anatomic due to children’s ability to remodel during the healing and recovery process, there are certain guidelines in the literature that define acceptable displacement with orthopedic management [1]. However, if acceptable reduction is not achieved, surgical management is mandatory, with a constant increase in indication in recent decades; Flynn et al. reported a “seven fold increase in operative treatment of these fractures”. This important increase in surgical treatment is due to technical advances (better quality of intra-operative imaging and lower dose radiation equipment, development of titanium elastic nails and pediatric-specific implants), clinical experience and a general intolerance to deformities both on the surgeon’s behalf, as well as from the patient’s family. When operative treatment is decided upon, intramedullary fixation is the first choice because of the shorter anesthesia time, shorter hospital stay and faster healing [2]. Also, this type of fixation is achieved through minimally invasive approaches, small incisions and minimal soft tissue aggression, all of which translate to more rapid healing and recovery, as well as fewer local complications like infection or wound problems. If closed reduction is not successful, open reduction is necessary as well as K-wire fixation, achieving fracture reduction, stability and the restoration of normal anatomy [3]. Plate fixation is uncommon in pediatric surgery, but can be used with care in adolescents and in particular cases like with the presence of complications such as non-unions, where absolute stability is necessary to achieve bone healing. The use of plate fixations needs special attention regarding plate placement and growth processes and biology.

Post-traumatic pediatric non-unions are rare, mainly because of increased local biology in young patients, with significant evidence suggesting that the periosteal sleeve plays an essential role in healing [4]. The potential factors described in the literature for failure are neglect, unstable fixation, early removal of implants, improper follow-up management or a combination of these [5]. With proper clinical management, non-unions should not occur in children.

The present paper aims to propose another unique non-union-causing complication that should be taken into consideration when the above-mentioned factors have been excluded, and this represents a unique finding in a case of non-union of the distal radius, namely the traumatic volar transposition of the extensor carpi radialis longus tendon. This has led to repeated displacement during conservative attempts, and the chronic non-union of the fracture site after closed reduction and percutaneous fixation using K-wires. A continuous tendency of the distal fragment to angulate anteriorly and a failure to heal can lead to open surgery and a constant sense of incomplete understanding of the pathophysiology on the surgical team’s behalf. Ultimately, the search for reasons will continue during the operation and close attention to soft tissue anatomy has revealed the answers to understanding this unique complication.

## 2. Detailed Case Description

A 10-year-old child presented at the pediatric emergency department with acute pain and functional disability of the right forearm and hand, following a fall while running. Patient BMI (body mass index) was normal and the general activity level was high; the child had been undergoing regular handball training for two years before the traumatic event. On presentation, the clinical exam revealed a slight swelling of the forearm and deformity of the distal third of the forearm with palmar angulation, with normal sensitivity and circulation distally and the ability to move fingers and thumb but with increased pain in the distal forearm. Blood tests and pre-anesthetic work-up were within physiological ranges, with no anterior pathologies to report.

A plain X-ray at initial presentation confirmed a fracture of the distal third of the radius with important displacement and volar angulation >30° (Figure 1A). At presentation, orthopedic reduction was the first gesture but this failed by orthopedic means with a non-successful conservative attempt.

Due to the fact that orthopedic maneuvers were not efficient, the patient was taken to the operating room and the maneuvers repeated under general anesthesia, but this was also inefficient. A mini open reduction technique was applied, in which a K-wire was introduced into the fracture site through a <1 cm skin incision and used as a lever to facilitate reduction. Once reduction was obtained under fluoroscopy, the fracture was fixed percutaneously using two crossed K-wires, obtaining satisfactory stability and the final result (Figure 2).

Post-operatively, the patient was immobilized in a cast splint for dressing protection, with gravitational drainage with the forearm in a sling, and presented for follow-up at two and four weeks. At 6 weeks, he was admitted for K-wire removal. However, after the removal of the K-wires the forearm became painful over the next few days, with progressive deformity of the distal forearm, which sent the patient back to the hospital in one week, and a radiological control exam was obtained. The fracture appeared with slight callus formation and volar angulation of the radius fracture. Because of the radiological aspect suggesting callus formation, the patient underwent soft orthopedic manipulation with reduction and was placed in a splint for four weeks. At the end of the 4-week immobilization, the splint was removed and a radiological exam obtained (Figure 3). At this point, 10 degrees of volar angulation was considered acceptable and the patient was advised to mobilize the hand and wrist as tolerated. Bear in mind that here we were situated in week 11 in this case’s chronology. Recovery did not develop as expected as there was intermittent pain with movement, a lack of mobility progress and progressive recurrence of the distal forearm deformity.

Follow-up exams were scheduled monthly, but the patient did not present until 6 months post-operatively, with pain over the forearm, increased pain with movement of the radiocarpal joint and thumb, moderate loss of range of motion and objective deformity of the distal forearm with no history of recent trauma. A plain radiograph was obtained (Figure 4).

Reconsidering the evolution and the clinical outcome, open surgical reduction and stable fixation became mandatory. Under general anesthesia, a dorsal approach was elected, partly to respect the initial post-operative scar, and a distal forearm locking plate was selected beforehand. Intra-operatively, after complete dissection and the exposure of the fracture site, the cause for continuous volar displacement was evident, and this was a traumatic transposition of the extensor carpi radialis longus tendon to the volar surface of the radius through the fracture site (Figure 5A). Intra-operatively, the extensor tendon was wrapped around the radial shaft, leading to volar angulation and fracture site movement, leading to non-union and the failure of the radius to heal (Figure 3).

After identifying the tendon and the anatomical position, this was reduced and passed dorsally through the fracture site, retracted, and fixation was obtained under fluoroscopy using a locking plate, located proximally to the growth plate, with satisfactory reduction and absolute stability (Figure 5).

Post-operatively, no immobilization was required and mobility was allowed as tolerated without loading the hand for four weeks. At this time, the passive range of motion was normal while the active range of motion was slightly limited for flexion and extension of the wrist, and the patient underwent a 4-week rehabilitation program involving physical therapy exercises and mobility exercises. At 6 weeks post-operation our patient was pain-free with full range of motion. At week 8 the patient started strengthening exercises, and was allowed to return to high-demand sports after the 4-month follow-up clinical evaluation, which revealed full range of motion, passive and active, and no pain when loading the hand during active flexion and extension. At 12 months, we obtained a radiological control image (Figure 6) that confirmed healing and adequate callus formation as well as bone remodeling.

The final result was adequate healing of this malunion, which was not possible until the tendon of the extensor carpi radialis longus had been repositioned dorsally in the anatomical position. This finding is unique when reviewing the current literature and might be an important cause of malunion in children, especially in fractures with significant traumatic displacement and exaggerated fracture site angulation at the moment of trauma, or in cases where orthopedic maneuvers fail to obtain adequate reduction of the fracture site.

A consistent search for causal factors and a lack of medical explanation for this complication in this particular case were the main reasons why the surgical team was extremely meticulous with the dissection and intra-operative observation of the forearm anatomy. A lack of medical understanding might be key in continuously searching for answers, and in this situation led to an extremely interesting intra-operative finding.

## 3. Discussion

Tendon interpositions in fracture sites or even dislocations have been described in many instances as local complications that lead to reduction failure, like the long head of the biceps tendon in proximal humerus fractures [6,7,8,9], phalanx fractures or dislocations complicated by flexor tendon interposition [10], extensor digitorum interposition in radius fracture [11], and other examples described, but there is no case documented in the literature that we could find describing a complete transposition through the fracture site. This only amplifies the suspicion that this might be easily overlooked when functional impairment is not significant. Raet et al. reports a case of distal radius non-union after plate fixation in a 9-year-old [12] where the radiological images appear similar to ours considering a pattern of recurrent displacement. Even in the 6-month follow-up image demonstrating nearly complete consolidation, one can still observe the tendency to angulate anteriorly. Could it be justified to question whether the cause of non-union and final “nearly complete consolidation” with residual volar angulation was in fact a traumatic tendon transposition?

After recognizing this unique surgical finding, all possibilities were taken into consideration regarding the moment when the palmar reposition occurred. From these possibilities, the authors mention two clinically relevant moments: after starting mobilization (this was after 6 weeks, so chances are practically inexistent) and during orthopedic maneuvers (repeated traction and reduction maneuvers might cause such tendon displacements, but reduction techniques were managed by experienced pediatric surgeons using gentle mobilization movements without applying the extreme forces needed for tendon dislocations). Taking this into consideration, and also the fact that correct reduction was simply not possible by orthopedic means (which is extremely rare in children) in this distal radius fracture, represent solid arguments that this complication in fact occurred at the moment of trauma due to gross displacement in the moment of the fall.

This type of complication should be considered in cases of closed fractures that evolve towards non-union or malunion with no apparent reason. Also, a continuous tendency to displace or angulate in exactly the same position as the initial displacement can also indicate traumatic tendon transposition. Meticulous functional examination by isolating muscles and testing them clinically and individually should be part of the forearm fracture clinical exam in order to diagnose pathological transpositions, but in the pediatric setting this is not always possible, and pain can also affect the accuracy of the examination.

A lack of documented cases of this particular complication translates to the great surgical value of our case report, and most probably this kind of complication has been overlooked in the past, if we consider that most non-unions are operated on later in the clinical time frame. This delay could make the anatomy of the surgical approach far more complicated and abnormalities like this one more difficult to diagnose intra-operatively. Also, the general surgical tendency to use smaller incisions could impair the correct visualization and understanding of the soft tissue anatomy, which leads to clear indications for wide-open approaches when such a situation is suspected.

In specific cases where soft tissue complications are suspected, soft tissue sonography could be extremely helpful in diagnosing tendon transpositions. Magnetic resonance imaging could also diagnose such a complication. When comparing the two imagistic investigations, the latter is significantly more expensive and not always available. However, in both methods, the most important aspect is the experience of the clinician performing image interpretation and their ability to identify imagistic anatomy. Just like the famous proverb, “beauty is in the eye of the beholder”, in both imagistic techniques identifying such soft tissue complications would not be easy, nor is it easy to recognize this during surgery.

## 4. Conclusions

Traumatic tendon transpositions through the fracture site should be included in the differential diagnosis of closed fractures with important displacement and repetitive displacement that tend to fail in healing.

When there is such a suspicion that MSK (musculo-skeletal) sonography or MRI imaging could be useful in confirming this complication and could be included in the pre-operative planning, more importantly, wide surgical exposure with an appropriate approach is essential in recognizing such a complication intra-operatively.

Fracture healing is the ultimate goal in pediatric trauma, but surgical management should always respect anatomy restoration principles without disregarding soft tissues in this noble quest.

## Figures and Tables

**Figure 1 children-11-00116-f001:**
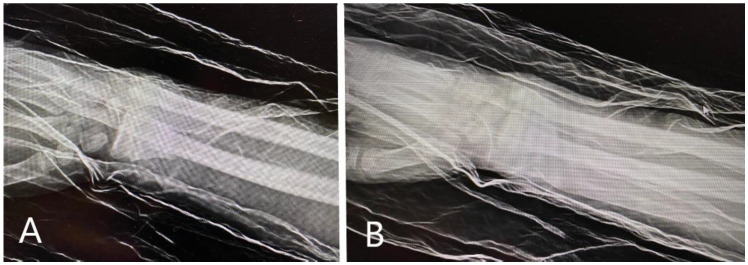
Radiological images obtained at presentation: AP (antero-posterior) (**A**) and oblique (**B**) view of the distal forearm, demonstrating important volar displacement of the radial fracture, and greenstick fracture of the distal ulna with dorsal angulation. Also, the fracture presented with a rotational displacement of the distal radial fragment together with the radiocarpal complex.

**Figure 2 children-11-00116-f002:**
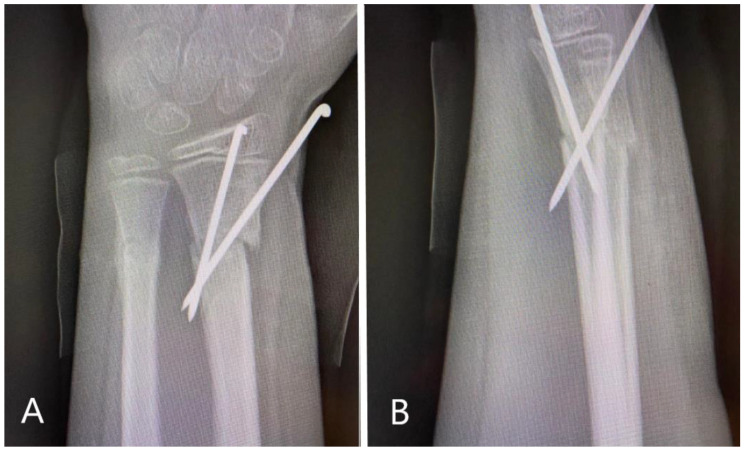
Radiological images obtained post-operatively. (**A**): AP view of the distal forearm, demonstrating good reduction of the fractures and fixation of the radius with two crossing K-wires according to AO pediatric technique. (**B**): Lateral view with very good reduction of the volar angulation of the radius, with slight but acceptable displacement of the ulna.

**Figure 3 children-11-00116-f003:**
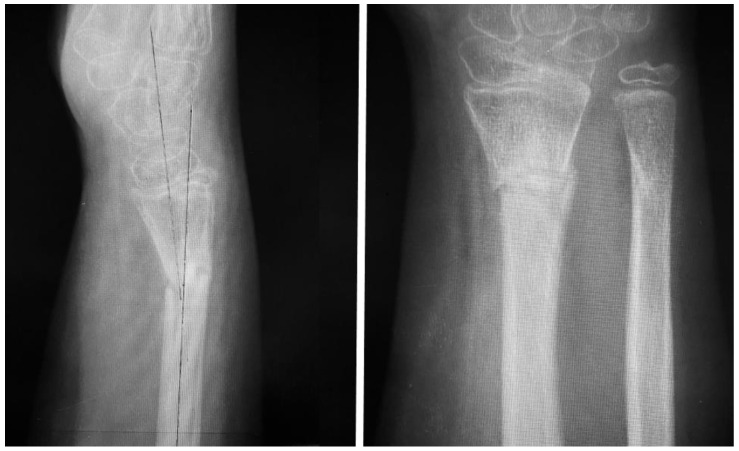
AP and lateral views obtained at the end of four weeks’ immobilization, demonstrating 10 degrees of volar angulation, which was considered acceptable at this point, but with inadequate callus formation.

**Figure 4 children-11-00116-f004:**
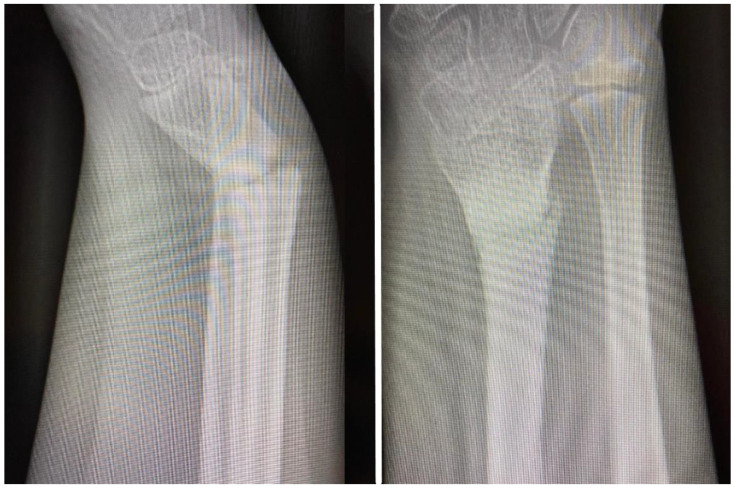
AP and lateral views obtained at 6 months after rehabilitation failure and recurrent pain with movement, demonstrating a lack of union and gross angulation of the radius and good healing of the ulna.

**Figure 5 children-11-00116-f005:**
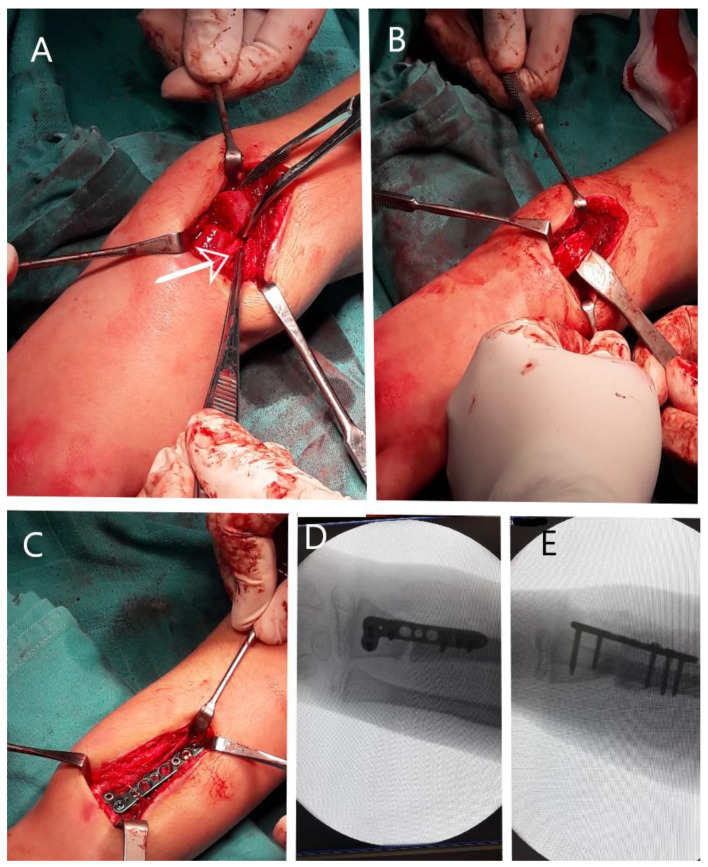
Intra-operative images: (**A**) intra-operative aspect with arrow pointing at the extensor carpi radialis longus tendon malpositioned on the volar side of the fracture site, in a spiral fashion around the proximal radius; (**B**) tendon of the extensor carpi radialis longus muscle reduced by passing the tendon through the fracture site to the dorsal side of the radius; (**C**) intra-operative aspect of the final fixation with locking plate placed on the dorsal aspect of the radius, proximal to the growth plate; (**D**) intra-operative fluoroscopy AP image demonstrating satisfactory reduction of the fracture and final fixation; and (**E**) fluoroscopy image in lateral view demonstrating adequate reduction and fixation as well as screw length.

**Figure 6 children-11-00116-f006:**
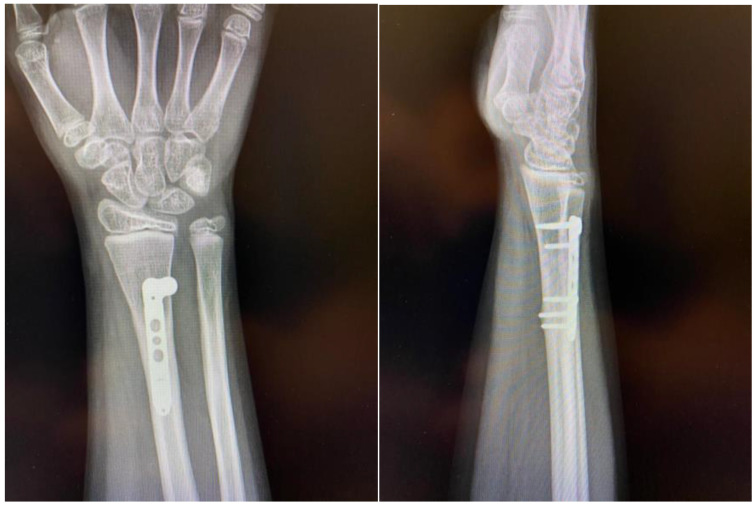
The 12-month follow-up plain radiographs demonstrating good healing and proper callus formation.

## Data Availability

The original contributions presented in the study are included in the article, further inquiries can be directed to the corresponding author.

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
