# Peer review of "Case Report of a Unique Intra-Operative Finding in a Pediatric Distal Radius Non-Union: Does This Shed Light on Other Non-Unions or Malunions in Children?"

_children, 2024, doi:10.3390/children11010116_

Round 1

Reviewer 1 Report

Comments and Suggestions for Authors

The authors describe a case report introducing a traumatic tendon reposition through the fracture site of a distal radius fracture. Although tendon repositions have been described as causes for non-union at other locations before, according to the authors, this is the first report of this finding at a distal radius fracture.

The manuscript is well written and there are only minor comments:

Please consider shortening the heading.

E.g. "Case report of a tendon transposition resulting in a non-union of a distal radius fracture in a child."

Please change the manuscript according to the CARE guidelines for case reports in order to increase its scientific standard. This is a must in order for me to accept the manuscript.

Lines 41 quotation missing (Flynn et al.?)

Sometimes space before citation, sometimes not. Please use a space before a citation everywhere.

Please explain abbreviations at first mentioning. E.g. “OR” line 81., e.g. “AP” line 109 and “MSK” line 202.

Spelling: “Reconsidering” lines 122, “useful” line 202.

Please consider of including the results part in the part 2 “case description”.

Please change “that we could find” into “that the authors could find” line 174.

Comments on the Quality of English Language

Minor corrections needed as stated.

Author Response

Dear Reviewer,

First of all thankyou for your help and valuable observations!

We have shortened the title but still kept the question as we feel that it raises more interest in the readers. We have included the words case report in the title according to the CARE guidelines.

We used the Care guidelines as backbone but still also kept the MDPI format as suggested in the section for authors. Also some aspects of care guidelines are not aplicable, for example "patient perspective" when discussing a ten year old. We added more information about our patient and relevant data.

We made all the corrections sugested and explained the abreviations.

We included the results part in the case description as suggested.

With utmost respect

Adrian Lazarescu

Reviewer 2 Report

Comments and Suggestions for Authors

Clarification on Surgical Trends:

The article mentions a seven-fold increase in operative treatment of forearm fractures, attributed to technical advances and clinical experience. Can you provide specific examples of these technical advances and how they have contributed to the increased surgical interventions?

Justification for Intramedullary Fixation:

The preference for intramedullary fixation is mentioned, citing benefits such as shorter anesthesia time and faster healing. Could you elaborate on the specific advantages of intramedullary fixation over other methods, considering the pediatric population?

Detailed Case Presentation:

The case presentation is detailed and informative. However, could you provide more information on the decision-making process leading to the mini-open reduction technique, such as why it was chosen over other methods and its success rates in similar cases?

Rehabilitation Protocol:

The patient's return to high-demanding sports after four months is highlighted. Can you provide more details on the rehabilitation program, including specific exercises, milestones, and criteria for allowing high-demanding sports?

Long-Term Follow-up:

The long-term follow-up at 12 months shows satisfactory healing. Could you discuss any potential complications or challenges that might arise in the extended postoperative period, and how they were addressed in this case?

Discussion on Tendon Transposition:

The discussion mentions the uniqueness of a complete transposition of the extensor carpi radialis longus tendon through the fracture site. Can you elaborate on the potential implications of this complication and its relevance to clinical outcomes compared to other tendon interpositions?

Comparison with Existing Literature:

The article refers to Raet et.al.'s case, suggesting similarities in radiological images. Can you discuss any differences in the clinical presentations, treatment approaches, or outcomes between your case and the referenced case?

Potential Diagnostic Tools:

The conclusion suggests using MSK sonography or MRI imaging in cases of suspected tendon transposition. Could you discuss the availability, feasibility, and potential challenges associated with these diagnostic tools in a pediatric population?

Generalization of Findings:

The conclusion emphasizes the importance of considering traumatic tendon transpositions in closed fractures. Are there any limitations or specific criteria that might restrict the generalization of this finding to all closed fractures in the pediatric population?

Surgical Approach Considerations:

The discussion mentions the surgical tendency to use smaller incisions. Could you discuss how the choice of incision size might impact the ability to diagnose and address complications like tendon transpositions intra-operatively?

Author Response

Dear Reviewer, 

Thankyou for your time and valuable observations, we have taken all of them into consideration as follows:

-we have detailed the reasons for increase in surgical managed cases

-we have stated the advantages of intramedulary fixation in the pediatric population

-we have detailed the rehabilitation cronology and protocol

-we did not encounter long term challenges but still mentioned potential complications

-in the case of Raet et al. the last images presented do not demonstrate bone healing so we cannot discuss their final result, but even in the final images the fracture tends to angulate anteriorly after plate fixation which suggests they might have had the same problem. Beeing that it is pure speculation we cannot assume this so it was just a comparisson of similaeities between the two cases 

-we have included an entire paragraph in the discussion section about sonography and MRI

-regarding generalisation of findings we did specify that it should be suspected in cases with recurrent displacement to the initial position, impossible closed reduction, recurrent difformity, we did not suggest it should be suspected in all closed fractures of the radius

-we did detail the importance of wide open aproaches when suspecting such soft tissue complications

With utmost respect

Many thanks for your interest in our work and the good ideas and observations

Much apreciated

Adrian Lazarescu